# Delineating Fire-Hazardous Areas and Fire-Induced Patterns Based on Visible Infrared Imaging Radiometer Suite (VIIRS) Active Fires in Northeast China

Wenjun Li [1,2,†] , Peng Li [1,2,*,†] and Zhiming Feng [1,2,3]

1   Institute of Geographic Sciences and Natural Resources Research, Chinese Academy of Sciences, Beijing 100101, China
2   University of Chinese Academy of Sciences, Beijing 100049, China
3   Key Laboratory of Carrying Capacity Assessment for Resource and Environment, Ministry of Natural Resources, Beijing 101149, China
*   Correspondence: lip@igsnrr.ac.cn
†   The author contributed equally to this work.

**Abstract:** (1) Background: Fire affects global agricultural and/or forest ecosystems with high biomass accumulation. However, the delineation of fire-hazardous areas based on satellite-derived active fire intensity is not well-studied. Therefore, examining the characteristics of fire occurrence and development plays an important role in zoning fire-hazardous areas and promoting fire management. (2) Methods: A fire intensity (FI) index was developed with Visible Infrared Imaging Radiometer Suite (VIIRS) active fires and then applied to identify fire-hazardous areas in Northeast China. Combined with terrain, land cover and net primary productivity (NPP), the spatial and temporal characteristics of active fire occurrence were consistently analyzed. Next, a conceptual decision tree model was constructed for delineating fire-induced patterns impacted by varied factors in Northeast China. (3) Results: The accidental, frequent, prone and high-incidence areas of active fires defined by the FI index accounted for 31.62%, 30.97%, 26.23% and 11.18%, respectively. More than 90% of active fires occurred in areas with altitude <350 m above sea level (asl), slope <3° and NPP between 2500–5000 kg·C/m². Similarly, about 75% occurred in cropland and forest. Then, four fire-induced conceptual patterns driven by different factors were classified, including the agricultural and forest active fire-induced patterns (i.e., the Agri-pattern and FRST pattern) with NPP ranging 2500–5000 kg·C/m², and two others related to settlements and unused land with an altitude <350 m asl. The Agri-pattern dominates in Northeast China because of agricultural straw burning. (4) Conclusions: Despite the national bans of open burning of straws, active fires due to agricultural production have occurred frequently in Northeast China in the last decade, followed by small and sporadic forest fires. The approach for defining fire-hazardous areas and varied fire occurrence patterns is of significance for fire management and risk prediction at continental to global scales.

**Keywords:** VIIRS active fire; fire-hazardous area; fire-induced patterns; spatial recognition; Northeast China

## 1. Introduction

Fires continue to occur under the dual attitudes of anthropogenic "use" and "prevention" [1,2]. In recent decades, global environmental changes have aroused concerns from the international community [3]. In particular, the continuous occurrence of large-scale forest fires directly or indirectly affects many aspects such as global atmospheric environment, ecosystem resilience and public health [4–6]. On the one hand, the invasion of wildfires into villages, towns or cities usually causes serious property damage and even threatens human health and life. On the other hand, periodic wildfires of low to moderate intensity also play an important role in the maintenance of ecosystem functions and the control

of invasive species [7,8]. Meanwhile, biomass burnings emit large amount of particulate matter and/or trace gases into the atmosphere and further cause serious problems to public health [9,10]. Therefore, the impacts of climate change and human activities on the occurrence and development of past, current and future fires have become one of the main research topics of fire research [11,12]. Among them, effects of fire occurrence have been studied at different spatial and temporal scales, including issues about how plants adapt to wildfires and how ecosystem landscapes interact with wildfires [13–15]. In addition, the fire mechanisms lay a solid foundation for revealing the characteristics of historical wildfires and reconstructing the relationship between wildfires and climate change and forest dynamics over long periods of time [16,17].

Fire-hazardous zoning (FHZ) divides fire-hazardous areas into different levels for hierarchical management. Scholars have used different models to explore forest fire-hazardous zoning at varied spatio-temporal scales based on meteorological data and fire frequency records [1,5]. Internationally, according to the International Wildland-Urban Interface Code, the distance to the structural location of different fire risk sources (such as gas stations and factories producing inflammables) is grouped into different levels of fire-hazardous zones [18]. Correspondingly, a series of fire protection standards for different hazardous areas have been formulated [19]. In particular, after the San Bernardino fire in 1980, public resource regulations were passed in California and a thematic map of areas' fire hazard severity was generated [20], including three levels of medium, high and extremely high. In addition, other factors such as topography, climate and population are also considered for the generation of fire hazard maps, and FHZ is conducted according to the roles and contribution degree of the factors [21]. However, there is still a lack of classification of fire-hazardous levels based on varied scenarios of fire occurrence over long periods, as well as their frequency and intensity. The occurrence of active fire provides information such as the accumulation of combustibles and the size of the fire, etc. Therefore, the spatial expression of fire intensity is of great significance to the formulation of fire management measures such as effective prevention in regional fire-hazardous areas.

At the same time, fire-induced patterns were analyzed in combination with climatic factors, with little consideration of net primary productivity (NPP) [1,6]. Among the nearly 40 policies and measures related to the revitalization of Northeast China implemented by the State Council of China, two of them refer to ecological and environmental protection measures. The contents involve forest protection and improvement of ecological barrier functions which place an emphasis on forest fire prevention [9,22]. By law, large-scale agricultural or forest fires have been banned, but this cost-effective way of clearing land in agricultural production is still practiced locally. For example, the incineration of crop residues within certain limits is allowed in the northeastern region of China. With Visible Infrared Imaging Radiometer Suite (VIIRS) active fire products, taking Northeast China as a case study area, a new fire intensity (FI) index was developed and applied to identify fire-hazardous areas. Firstly, the spatial and temporal characteristics of active fire occurrence were analyzed. Secondly, a conceptual decision tree model was constructed for the delineation of fire-induced patterns combined with terrain, land cover and NPP. The classification of fire-hazardous areas and varied fire occurrence patterns is of significance for fire management and forecasting in Northeast China and may provide methodological guidance for continental to global analysis in this field. Furthermore, this research has practical significance for warning fire-prone areas and changing land management methods to mitigate negative changes in regional ecological environment.

## 2. Materials and Methods

### 2.1. Details of the Study Area

Northeast China, between 38°N~53°N and 118°E~134°E, includes three provinces of Heilongjiang, Jilin and Liaoning, referred to as the three provinces for short (Figure 1). It is located at the eastern end of the Eurasian continent, adjacent to Siberia in the north and the Bohai Sea and Yellow Sea in the south. The terrain is dominated by plains (e.g.,

Songliao Plain) and low–medium mountains. Among them, the Changbai Mountains and Xing'an Mountains are the most important natural barriers to the climate and terrestrial ecosystem in Northeast China. It also contains the Songhua, Nen, Wusuli and other big rivers. The Songliao Plain—comprising the Sanjiang Plain in the east, the Songnen Plain in the middle and the Liaohe Plain in the south—has fertile soil and deep soil layers. The main rivers and the formation of plains have huge economic and ecological value. The vast flat and fertile land of Northeast China provides unique conditions for the development of agriculture, animal husbandry and fisheries, and serves as a very important national food production base. The forest area is 38.75 million hectares, accounting for 14.7% of China, with a forest coverage rate of 39.6% (Statistical Bulletin 2019). Thus, Northeast China has rich forestry materials including important timber and wild animal and plant resources. Various ecological reserves in Northeast China are important carriers of forest belts in China's ecological pattern of "two screens and three belts".

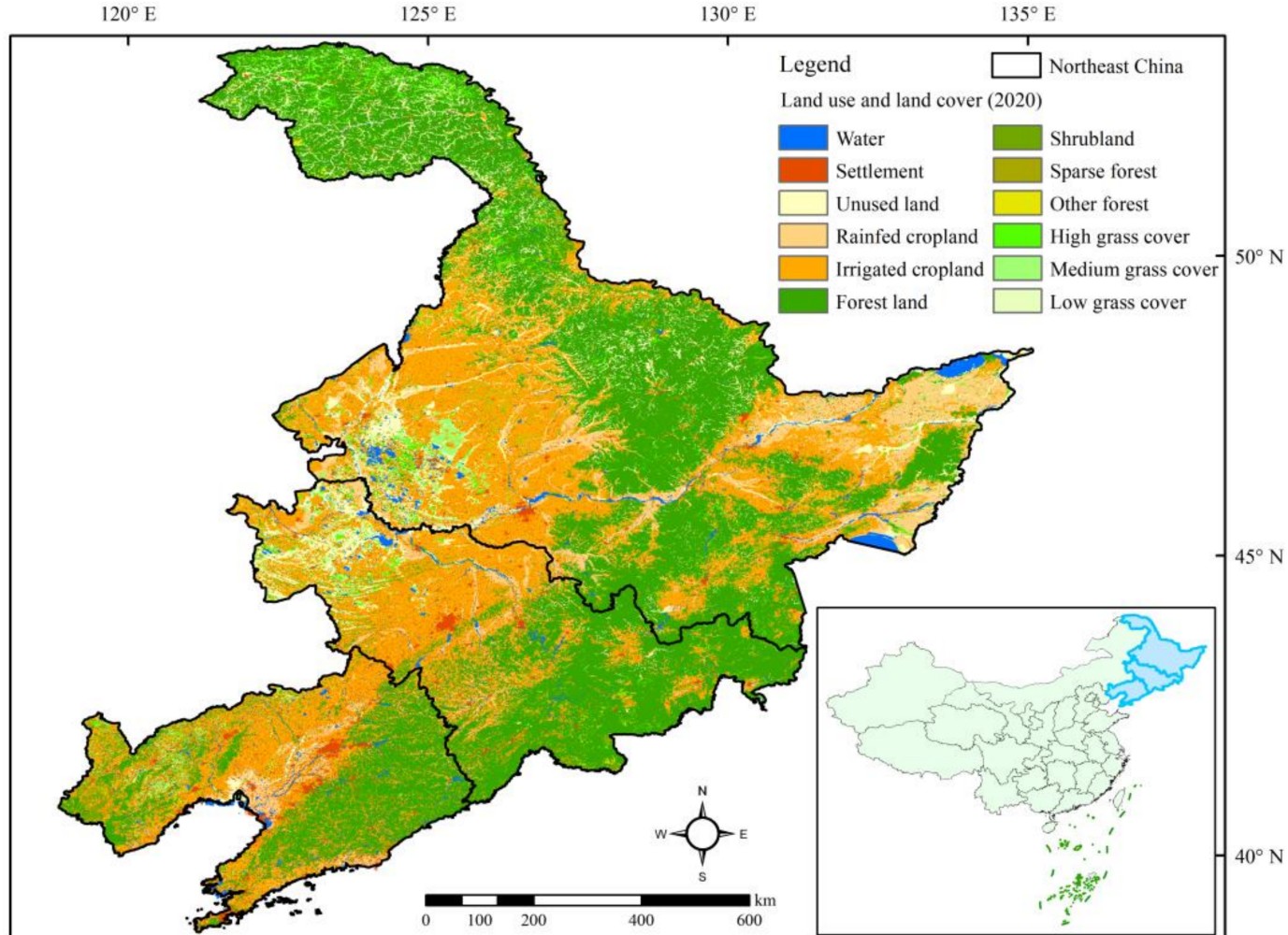

**Figure 1.** Maps showing the location of Northeast China and the 2020 land cover types.

Northeast China has a temperate monsoon climate, including the middle temperate zone and the cold temperate zone from south to north. It has four distinct seasons, with warm and rainy conditions in summer and cold and dry conditions in winter. The average annual temperature is below 0 °C, and the maximum reaches over 40 °C. Annual average precipitation varies greatly at different spatial and temporal scales ranging between 430.4 mm and 678.72 mm. In the past century, there has been a noticeable trend of temperature increase, while the annual precipitation has shown a decreasing trend [23]. With large

climate variability and frequent meteorological disasters, it is a typically vulnerable area suffering from vegetation fires.

*2.2. Materials*

2.2.1. Visible Infrared Imaging Radiometer Suite (VIIRS) Active Fire Data

The Fire Information Resource Management System (FIRMS), jointly developed by the National Aeronautics and Space Administration (NASA) Applied Science Program and the Food and Agriculture Organization (FAO) of the United Nations, released the Visible Infrared Imaging Radiometer Suite (VIIRS) version 1 active fire dataset. The data format includes shapefiles (.shp), comma-separated text files (.csv) or JSON files (.json). Shapefile data include information such as latitude and longitude of the corresponding grid center point, acquisition time, brightness, fire radiation power (FRP), collection azimuth and sensor information, etc. The active fire point represents the pixel center of 375 m × 375 m, and the study period is 2012–2020.

2.2.2. Global Digital Elevation Model (GDEM) Data

The Advanced Spaceborne Thermal Emission and Reflection Radiometer (ASTER) Global Digital Elevation Model (GDEM) version 2.0 data products, covering the entire study area, were freely gathered from the websites of the Japan Space Agency (JAXA, http://gdem.ersdac.jspacesystems.or.jp/ (accessed on 18 October 2020)) and the National Aeronautics and Space Administration (NASA, http://reverb.echo.nasa.gov/reverb/ (accessed on 26 October 2020)). Each compressed datapoint downloaded contains two files, namely the digital elevation model (DEM, dem.tiff) data and quality control (QA, num.tiff). The coordinate system is WGS84/EGM96, with a spatial resolution of 30 m.

2.2.3. National Land Cover Type Products

National land cover type products in Northeast China in 2020 were gathered from the Resource and Environment Science and Data Center by the Institute of Geographic Sciences and Natural Resources Research, Chinese Academy of Sciences (http://www.resdc.cn/Datalist1.aspx?FieldTyepID=1,3 (accessed on 13 October 2021)). The 2020 land cover data products were updated via artificial visual interpretation with Landsat 8 Operational Land Imager images, mostly based on the land cover remote sensing classification results generated in 2015. Land cover products include six level-1 types of cropland, forest, grassland, water body, settlement and unused land, as well as 25 s-level types (Table 1).

2.2.4. Net Primary Productivity (NPP) Data

The NPP data are those of NASA's Moderate Resolution Imaging Spectroradiometer (MODIS) C6 MOD17A3HGF.006, or the Terra Net Primary Production Gap-Filled Yearly Global product, with the unit of kg·C/m$^2$ and a spatial resolution of 500 m. To meet the temporal requirements of the study period with VIIRS active fire products, annual NPP data products during 2012–2020 were gathered and mosaicked via ArcGIS and then applied to calculate the NPP of agricultural and forest areas.

2.2.5. China's Ecological Function Reserve (EFR)

The China's Ecological Function Reserve (EFR) data were provided by the Resource and Environmental Science Data Center of the Chinese Academy of Sciences (https://www.resdc.cn/data.aspx?DATAID=272, (accessed on 13 March 2022)). They cover the extent of 50 ecological function-protected areas in China. EFRs with different ecological functions play important roles in water and soil conservation, biodiversity conservation and flood regulation and storage, as well as in marine ecological function protection and species resource ecological function protection. Northeast China has about 33% covering by varied EFRs with different functions. There are four major EFRs including the Songnen Plain wetland EFR, the Sanjiang Plain Wetland ERF, the Changbai Mountain EFR and the Liaohe

River Delta Wetland EFR. The EFR data are used to evaluate the occurrence of large-scale and high-intensity active fires.

**Table 1.** The Classification System of the National Land Cover Types in 2020.

| CLASSES | Secondary Classification System | Value |
|---|---|---|
| 1. Cropland | Irrigated cropland | 11 |
| | Rainfed cropland | 12 |
| 2. Forest | Forest land | 21 |
| | Shrubland | 22 |
| | Sparse forest | 23 |
| | Other forest | 24 |
| 3. Grassland | High-cover grassland | 31 |
| | Medium-cover grassland | 32 |
| | Low-cover grassland | 33 |
| 4. Water body | Wetland, lake, etc. | 41/42/43/44/46 |
| 5. Settlement | Urban, rural residential area, etc. | 51/52/53 |
| 6. Unused land | Sandy land, bare soil, saline–alkali land, etc. | 61/62/63/64/65/66/67 |

### 2.3. Methods

Based on the occurrence frequency, fire radiation power and spatial location attributes of active fire data, a fire intensity (FI) index was constructed and then applied to quantitative classification and spatial visualization through GIS (Figure 2). At the same time, combined with terrain, land cover and NPP data, the spatial and temporal characteristics of active fire occurrence were analyzed. Next, we attempted to construct a conceptual decision tree model for examining fire-induced patterns in Northeast China. Finally, the spatial analysis of different induced patterns of active fires in Northeast China was finished accordingly.

#### 2.3.1. Point Density Analysis

The point density analysis tool can be used to calculate the density of point features around each output raster cell. Conceptually, a neighborhood is defined around the center of each raster cell, and the number of points in the neighborhood is summed and divided by the neighborhood area to derive the density of point features. If a value of each item (other than NONE) is used for the population field setting, it will be used to determine how many times the point is counted. Calculation of the magnitude within each unit area is generally based on the point features that fall into the neighborhood around each unit. The larger the value of the input radius, the higher the generalization degree of the generated density grid will be, and vice versa. When calculating the density, only the points that fall within the neighborhood are considered. If no point falls within the neighborhood of a specific cell, then NoData is assigned. In this study, the spatial resolution of the output raster was 500 m, with the search shape chosen as circle. By contrast, the search radius was 100 active fire "points" centered by a given point. It should be noted that one raster represents multiple active fire events that occurred on the same site in different time periods. Finally, the point density analysis tool was applied to conduct grid spatialization of active fire occurrence frequency and FRP.

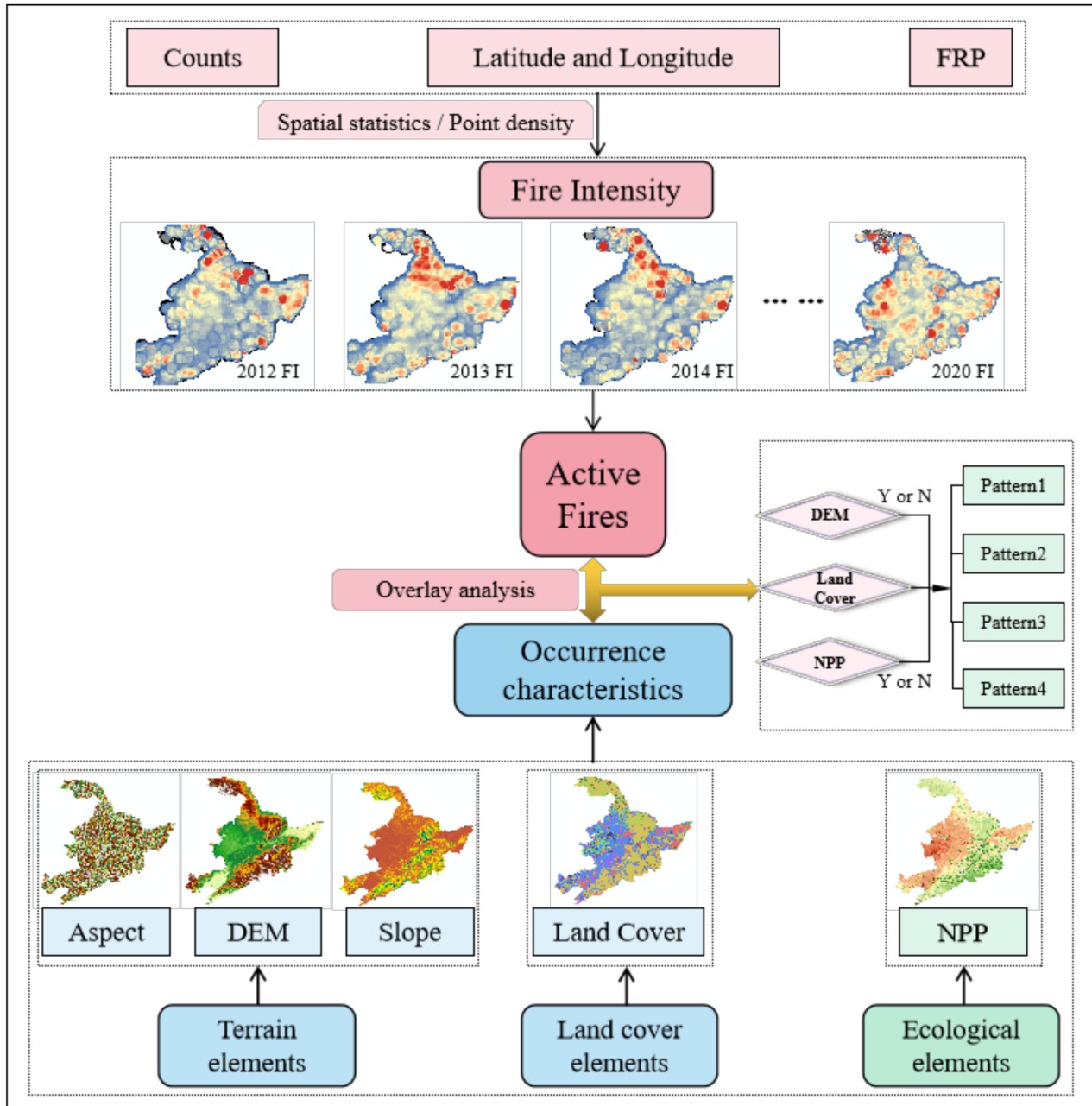

**Figure 2.** The workflow for delineating fire-hazardous areas and fire-induced patterns based on Visible Infrared Imaging Radiometer Suite (VIIRS) active fires in Northeast China.

2.3.2. The Construction of the Fire Intensity Index

Based on attributes such as the location, time and FRP of active fire events, the characteristics of active fires can also be revealed through statistical methods. Although FRP and other attributes of an active fire "spot" can be spatially explicit, they cannot effectively depict the combined effects of FRP and fire frequency simultaneously in the same site. At the same time, the total fire radiation frequency in certain geographical spaces is high due to two major reasons. First, there are active fire events with very small FRP. Second, very few or even one large fire occurred in a certain area resulting in a high value of FRP. For this reason, the FI index is established to overcome the situation wherein the local total FRP is large due to the occurrence of multiple active fire events. Based on GIS spatial

analysis, the FI index aims to spatialize the frequency of active fires (including location information) and FRP simultaneously. The formular of the FI index is given as follows:

$$FI = FRP/FD \tag{1}$$

where FI is the fire intensity index, FRP is the gridded fire radiation power and FD is the gridded fire density. After calculation, the FI index normally fluctuates between 0 and 50 in Northeast China with the exception of a maximum of up to 161 in 2013. The occurrence intensity of annual active fires is classified into four grades using the natural breakpoint method based on the FI index, corresponding to the class one, two, three and four or the accidental, frequent, prone and high-incidence Areas, respectively.

## 3. Results

### *3.1. Spatial Development of Active Fires in Northeast China from 2012 to 2020*

#### 3.1.1. Spatial Characteristics of Active Fires

The frequency and spatial characteristics of active fires in Northeast China have changed significantly from 2012 to 2020, showing a fluctuating trend. There were several annual peaks in 2014, 2017 and 2020, with occurrence frequencies of $19.49 \times 10^4$, $20.05 \times 10^4$, and $22.15 \times 10^4$, respectively. However, the frequency of active fires in the first one to two years before reaching the peak is extremely small, namely in 2012 ($5.99 \times 10^4$), 2016 ($12.23 \times 10^4$) and 2018 ($7.40 \times 10^4$). The occurrence of active fires in Northeast China is distributed in the southwest of the Sanjiang Plain, the east of the Songnen Plain, and the Liaohe Plain. Specifically, the density of active fires in the Liaohe Plain has maintained a high value from 2012 to 2020. Next, the high-value areas of annual active fire point density in the Sanjiang Plain spatially gathered in clusters from 2012 to 2015. Since then, the areas of high-value active fires have been scattered geographically. In contrast, the spatial development process of active fire density in the Songnen Plain was more complicated. It takes a three-year cycle and presents a cyclical development characteristic of multi-point dispersion and high-value agglomeration (Figure 3).

#### 3.1.2. Spatial Characteristics of Active Fire Radiation Power

Differing from the occurrence frequency, the FRP represents the magnitude of the FI of an active fire. The FRP value of each active fire vector point is subjected to density interpolation, and the results are shown in Figure 4. On the whole, the FRP of active fires in Northeast China fluctuates and increases every three years. For example, the average FRP of active fires increased from 1056 MW in 2012 to 4255 MW in 2014. After that, the average FRP of active fires reached high values in 2015 (4650 MW) and 2017 (4519 MW). The maximum FRP of a single active fire occurred in 2020, indicating that a single high-FRP active fire occurred during that year. In terms of spatial development characteristics of FRP of active fires in Northeast China, the areas with high FRP were concentrated in the western Sanjiang Plain, the Songnen Plain and the Liaohe Plain. Specifically, from 2012 to 2020, the active fire intensity in the Liaohe Plain continuously kept a high value. By 2014, the high-value area of FRP was concentrated in plains. From 2012 to 2017, high values of FRP in the Sanjiang Plain spatially gathered in clusters, and then gradually decreased and scattered during 2018–2020. The Songnen Plain is the area with the most extensive distribution of high-FRP active fires, with a three-year cycle, showing the cyclical development characteristics of multi-point dispersion and high-value agglomeration.

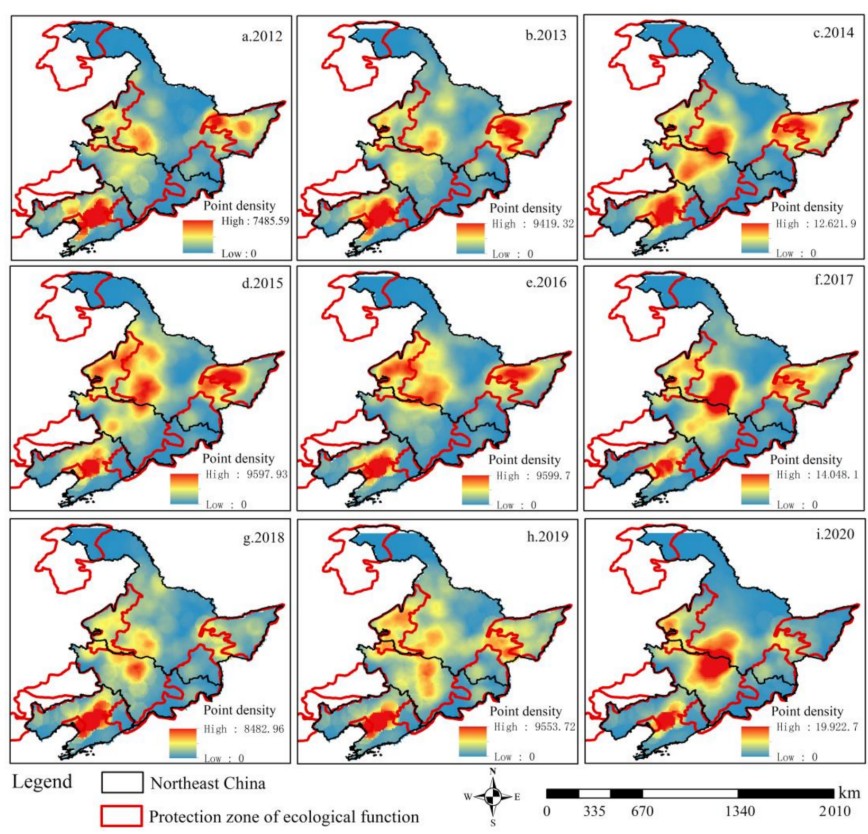

**Figure 3.** Annual point density of active fires in Northeast China from 2012 to 2020.

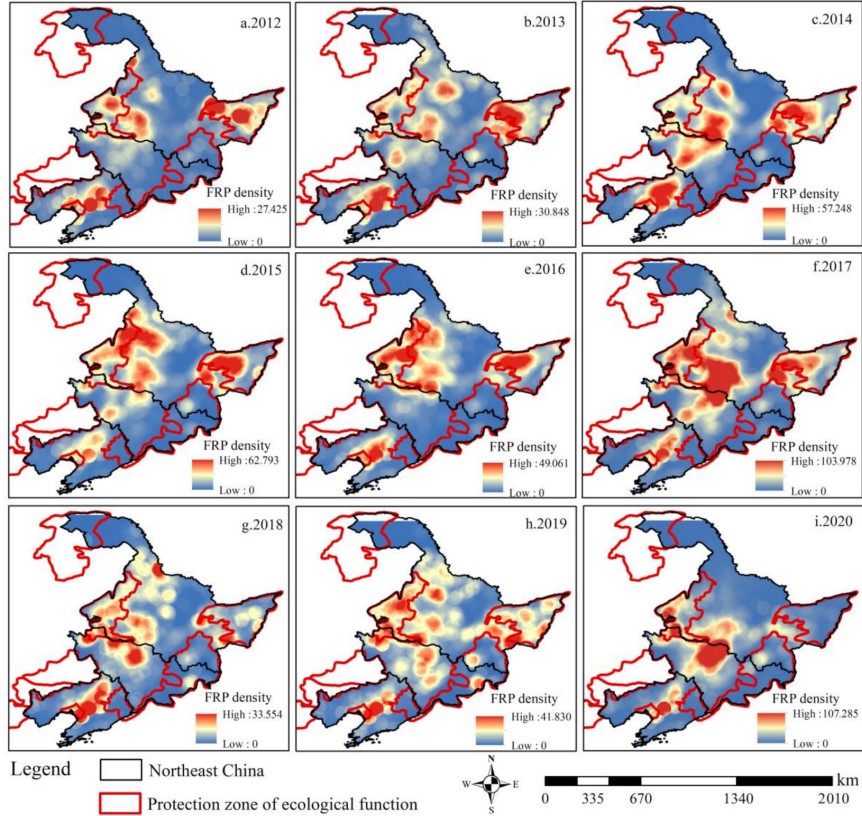

**Figure 4.** Spatial characteristics of fire radiation power (FRP) of VIIRS active fires in Northeast China during 2012–2020.

### 3.1.3. Spatial Identification of Varied Active Fire Intensity

The four classes of active fire intensity during 2012–2020 show noticeable spatial variations in Northeast China (Figure 5), with the high-incidence area corresponding to the level 4. Among them, the accidental area had 31.62% of active fires, followed by the frequent area, prone area and high-incidence area accounting for 30.97%, 26.23% and 11.18%, respectively. In particular, overlay analysis with the maps of China's Ecological Function Reserve (EFR) showed that 30.63% of the total active fires are located in EFRs. The land areas of accidental, frequent, prone and high-incidence areas in EFRs account for 32.68%, 27.38%, 31.59% and 8.35% of active fires, respectively. Among them, active fires occurred more frequently in the Songnen Plain, the Sanjiang Plain, the Changbai Mountain and the Liaohe Delta Ecological Reserve. Next, the four types of active fire intensity in Northeast China and the hazardous areas in Northeast China's EFRs as the key study areas were utilized to reveal the occurrence and development characteristics of active fires inside and outside the EFRs.

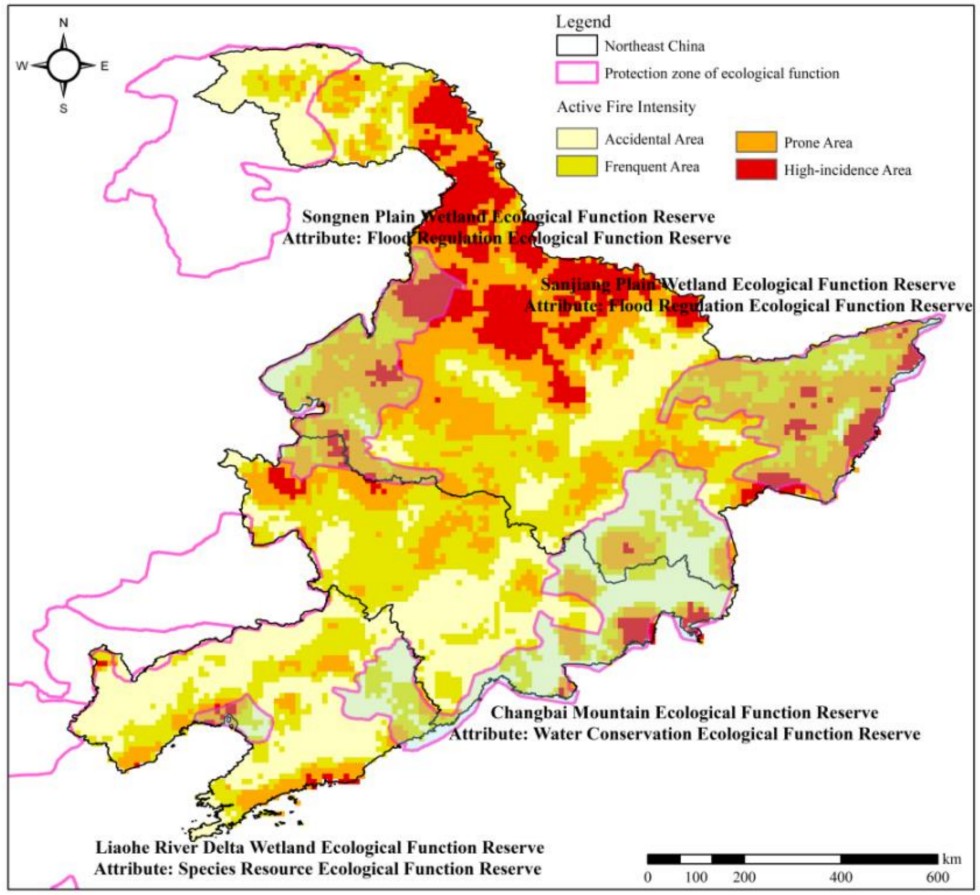

**Figure 5.** Classification map of fire-hazardous areas of VIIRS active fires in Northeast China.

### 3.2. *Characteristics of Occurrence and Development in Active Fire-Hazardous Areas*

3.2.1. Topographic Characteristics of Active Fire Occurrence and Development

Natural geography and man-made elements have a vital influence on the occurrence and development of active fires. Among them, topography affects the spatial characteristics of active fire occurrence, and land cover determines the strength of active fire susceptibility. Similarly, topography and land cover also impact the accessibility of human behavior to the occurrence of active fires. For example, local residents prefer to choose open plains with relatively small slopes for logging activities.

In general, more than 93% of active fires occurred at less than 350 m above sea level (asl) in Northeast China (Figure 6), and the frequency of active fires in the four types of

fire-hazardous area showed an obvious bimodal characteristic with the increase in altitude. Specifically, active fires in the accidental area mainly occurred in the plain areas of <80 m asl and 100–150 m asl, and the frequency of occurrence was 32.68% and 16.34%, respectively. The active fires in the frequent area are concentrated in the plain areas of 10–80 m asl and 100–230 m asl, and the corresponding occurrence frequencies are 18.99% and 66.64%, respectively. Similarly, the active fires in the prone area are also concentrated at 10~80 m asl and 110–210 m asl, with a corresponding frequency of 24.43% and 54.17%, respectively. In the high-incidence area, 13.25% of the active fires occurred below 80 m asl, and 27.11% of the active fires occurred at the altitude range of 110–170 m asl. Furthermore, there were still active fires in the hilly areas over 200 m asl.

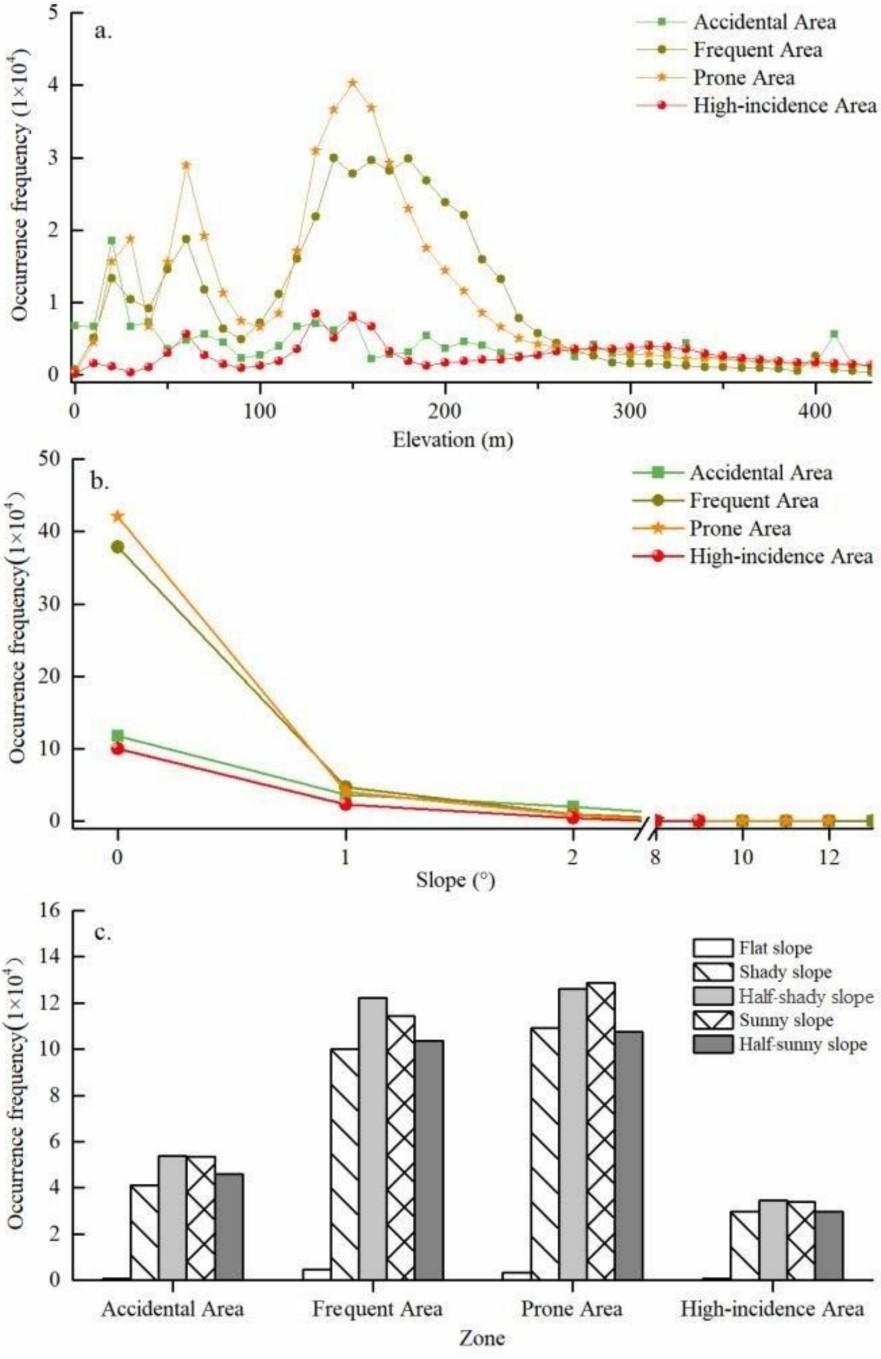

**Figure 6.** Topographic characteristics of fire occurrence frequency in active fire-hazardous areas in Northeast China. Note: (**a**) elevation, (**b**) slope and (**c**) aspect.

More than 90% of active fires in Northeast China occurred at a slope of less than 3°. In comparison, the quantity of active fire occurrence frequency is closely related to half-shady slopes and sunny slopes in the aspect distribution characteristics. The distribution characteristics of slope and aspect in the four types of active fire-hazardous areas show that: Active fires in the four types of fire-hazardous areas are highly concentrated on flat slopes, and the frequency of active fires gradually decreases with the increase in slope. About 80% of active fires in the frequent, prone and high-incidence areas are concentrated on flat slopes. In the above three types of areas, the probability of active fire occurrence is more than 98% on slopes <3°, with few and rare occurrence on slopes >10°. About 60% of active fires in the accidental area occurred on flat slopes, while the probability of active fires occurring on slopes <5° was 99%.

It should be noted that, affected by the relative order of magnitude of the frequency of active fires, differentiation of rising characteristics on the slope is weak, but the frequency of active fires on different slope gradients still varies from 1000 to 10,000 times. More than 53% of active fires in the four types of fire-hazardous areas occurred on half-shady and sunny slopes, and the proportion of active fires on shady slopes was the smallest. Specifically, the frequency of active fires on half-shady slopes and sunny slopes in the frequent areas reached $12.22 \times 10^4$ times and $11.45 \times 10^4$ times. Meanwhile, the frequency of active fires on half-shady and sunny slopes in the prone areas reached $12.60 \times 10^4$ times and $12.88 \times 10^4$ times, respectively.

### 3.2.2. Characteristics of Land Cover of Active Fire Occurrence and Development

Land cover carries combustibles and is the specific space where active fires occur. The overlay analysis of active fire-hazardous areas and land cover show that (Figure 7): more than 75% of active fires in Northeast China occurred in cropland and forests. In addition, 12.84%, 6.61% and 2.85% of active fires occurred in settlements, unused land and grassland, respectively. In terms of agricultural active fires, more than 52% of active fires occurred in rainfed cropland, followed by 14% in irrigated cropland. In the four types of is about 4:1. Among them, the frequency of active fires in rainfed cropland is the highest in the frequent and prone areas: $26.84 \times 10^4$ times and $25.25 \times 10^4$ times, respectively.

About 7% of active fires occurred in forestland in Northeast China, followed by less occurrence in shrubland (1.05%), sparse forest (0.37%) and other forest land (0.08%), respectively. In the four types of active fire-hazardous areas, forest land in the accidental area has the highest frequency of active fires, $2.91 \times 10^4$ times. In the frequent and prone areas, the frequency of active fires in forest land was about $2 \times 10^4$ times. Similarly, grassland also provides combustible substances for active fire burning, but the area of grassland in Northeast China is much small, and the resultant grassland fires are not common. Specifically, the fire frequencies of high-coverage grassland, medium-coverage grassland and low-coverage grassland in the four types of fire-hazardous areas were merely 1.67%, 1.09% and 0.08%, respectively. Among them, the counts of active fires in the middle- and high-coverage grasslands in the prone area were much larger, about $0.86 \times 10^4$ times and $0.71 \times 10^4$ times, respectively.

Compared with the landscapes such as cropland, forest and grassland, active fires in settlements and unused land usually have less natural vegetation as combustibles. Active fires in settlements are mostly closely related to human activities, e.g., urban fires. In addition to man-made transport of combustible materials for off-site incineration, active fires in unused land also do not exclude wildfires caused by natural factors, such as lightning strikes. Therefore, the active fires on impervious surfaces and unused land in Northeast China cannot be ignored because of large occurrence frequencies of 12.84% and 6.61%, respectively. At the same time, it was found that active fires in impervious areas are mostly concentrated in accidental, frequent and prone areas, and rarely occur in high-incidence areas. It can be seen that in Northeast China, the occurrence of fires in areas with frequent human activities had been strictly controlled in the past two decades.

In addition, the frequency of active fires on unused land in high-incidence areas and prone areas is relatively high, and the reasons behind this need to be further studied.

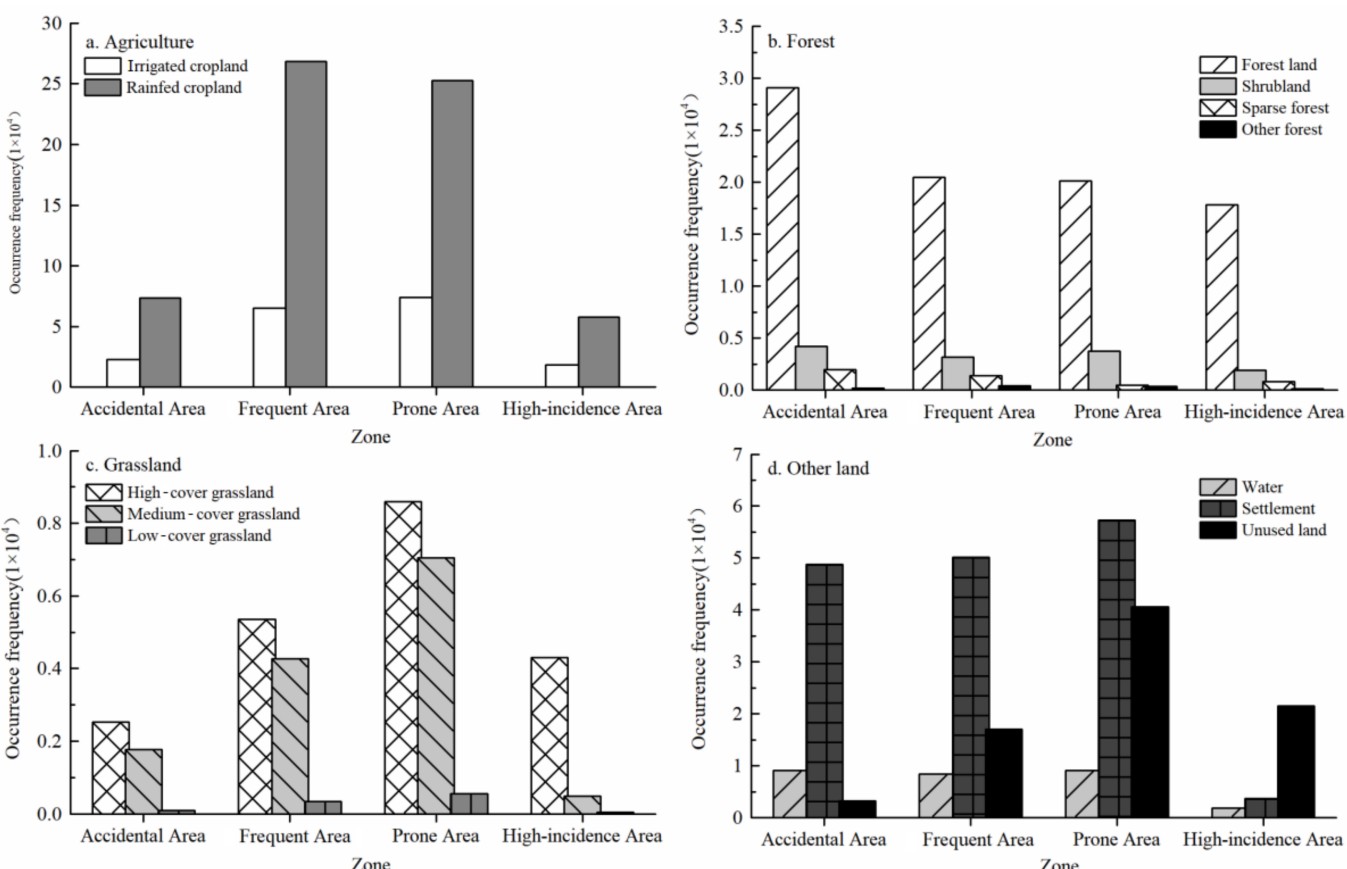

**Figure 7.** Occurrence characteristics of land cover in active fire-hazardous areas in Northeast China. Note: (**a**) Cropland, (**b**) Forest, (**c**) Grassland and (**d**) other land cover.

3.2.3. Characteristics of NPP of Active Fire Occurrence and Development

From 2012 to 2020, the average NPP in Northeast China also showed a fluctuating trend (Figure 8), and the average NPP peaked in 2014 and 2018, at 4721 kg·C/m$^2$ and 4696 kg·C/m$^2$, respectively. In addition, the average NPP changed drastically between years. In 2015, the mean value of NPP decreased to 4300 kg·C/m$^2$, which was the lowest during the study period. However, the mean value of NPP showed a steady trend after 2018. Combined with the 2012–2020 average NPP data, the correlation analysis of active fire occurrence frequency and NPP in different fire-hazardous areas was carried out in Northeast China.

In general, more than 93% of active fires occurred in the geographical space with NPP values ranging from 2500 to 5000 kg·C/m$^2$, and the frequency of active fires in the four types of hazardous areas was different from the variation characteristics of NPP. Specifically, the counts of active fires in the frequent and prone areas showed a trend of increase and then decrease with the increase in NPP. The frequency of active fires in the accidental area showed obvious fluctuations with the increase in NPP, and the frequency of active fires in the high-incidence area showed a bimodal change with the increase in NPP. In other words, more than 70% of active fires in the frequent and prone areas are concentrated in the geographical space with NPP values between 3000 and 4000 kg·C/m$^2$. As far as the high-incidence area of active fires is concerned, more than 80% of active fires occur in the geographical space with NPP values between 3000 and 5000 kg·C/m$^2$. This indicates that high-intensity active fires are more likely to occur in geographical spaces with high NPP values. However, active fires in the accidental area occur in regions with high and low

NPP values. And there is a weak correlation between the frequency of active fires and NPP. This shows that the occurrence probability and intensity of active fires in the accidental area are relatively random, which confirms the rationality of the above division of active fire-hazardous areas.

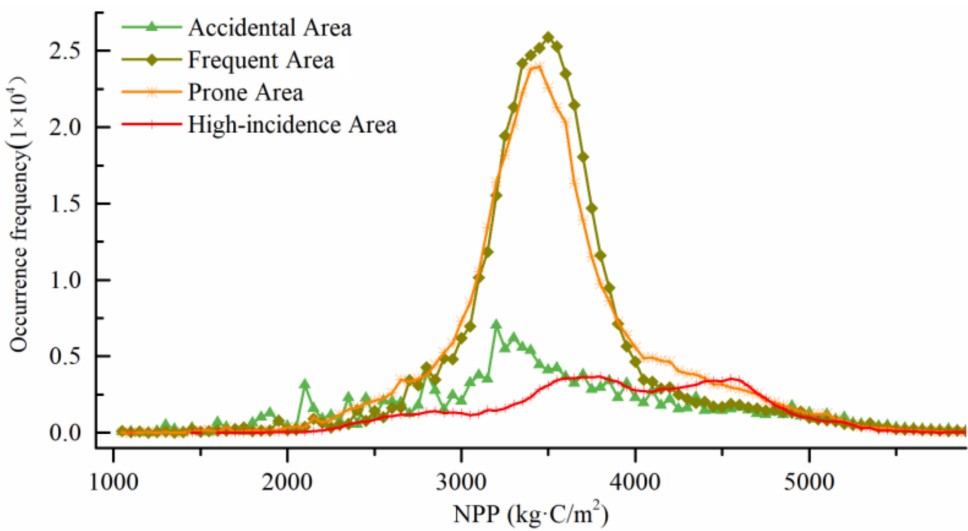

**Figure 8.** Characteristics of NPP in the four fire-hazardous area of active fires in Northeast China.

### 3.3. Active Fire Occurrence-Induced Concept Pattern Recognition

### 3.3.1. Active Fire-Induced Conceptual Pattern Construction

By analyzing the characteristics of active fires in different fire-hazardous spaces, characteristics of the terrain, land cover and NPP of active fires in Northeast China were fully examined. The results preliminarily show that cropland, forests, settlements and unused land are the four land cover types with the most counts of active fires. In addition, active fires occur most intensely in geographical spaces where the altitude is less than 350 m and the NPP is between 2500 and 5000 kg·C/m$^2$. Based on the above characteristics of active fire occurrence, a conceptual pattern of active fire occurrence induced by different factors is constructed (Figure 9).

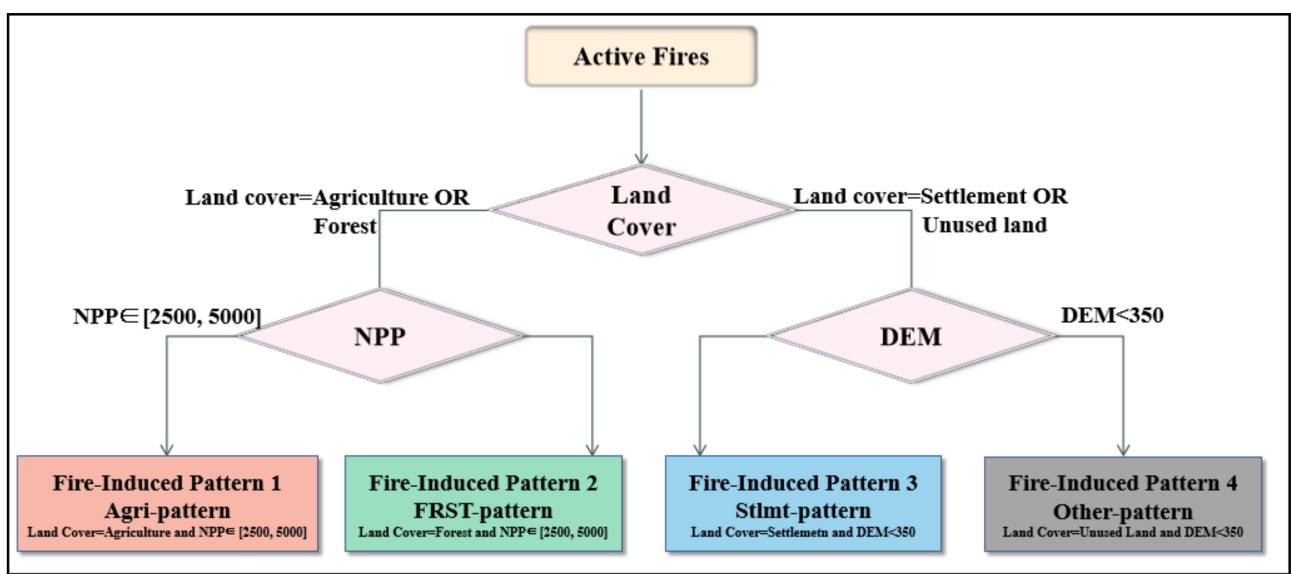

**Figure 9.** The decision tree based conceptual model for delineating fire-induced patterns based on VIIRS active fires in Northeast China.

This conceptual pattern defines four types of induced patterns of active fire occurrence. First of all, it should be noted that cropland and forests carry rich combustible biomasses, and it is necessary to devise the range of NPP values that are prone to active fires in geographical spaces with biomasses. Therefore, in the conceptual pattern, cropland with a NPP value between 2500 and 5000 kg·C/m$^2$ is defined as active fire-induced Agri-pattern, which represents an active fire-induced pattern of agricultural straw burning in cropland areas with high crop coverage. Similarly, forests with NPP values ranging from 2500 to 5000 kg·C/m$^2$ are defined as active fire-induced FRST-pattern, which represents an active fire-induced pattern of natural-induced wildfires in forests with high tree/shrub cover.

As the climatic and hydrological conditions vary slightly in Northeast China, topography is, thus, the most important natural geographical factor that determine land surface variations. At the same time, under the dual roles of land cover and topography, the type of unused land is also defined. Therefore, settlements with an altitude of <350 m are defined as active fire-induced Stlmt-pattern in the conceptual pattern, which represents man-made active fires caused by unnatural biomasses as combustibles in areas with frequent human activities. Unused land with an altitude of <350 m is defined as the active fire-induced other-pattern, which represents the active fire-induced pattern under the combined influence of human and natural factors. It is usually caused by anthropogenic burning of combustible substances in different places, while others are caused by some natural factors (e.g., lightning strikes).

### 3.3.2. Active Fire-Induced Pattern and Spatial Recognition

Based on the conceptual model mentioned above, the spatial features of different fire-induced patterns in Northeast China were examined through ArcGIS (Figure 10). The results show that the fire-induced Agri-pattern is the main pattern of active fire occurrence in Northeast China, normally seen in most regions including the Songnen Plain, the Sanjiang Plain, the Songliao Watershed and the Liaohe Plain. Because of the richness in soil and water resources and being one of the main crop-growing areas, the disposal of crop residues in Northeast China by incineration is allowed within certain limits. Therefore, the characteristics of active fire occurrence with man-made straw burning as the main fire-induced pattern are formed. At the same time, overlay analysis with maps of China's EFRs showed that there were still large-scale active fires in croplands in the the Songnen Plain Wetland, the Sanjiang Plain Wetland and the Liaohe Wetland Ecological Function Reserve.

In some areas such as the Xiao Hinggan Range, the Liaohe Plain and the Liaoning Hills, the main active fire mode is the fire-induced FRST-pattern (Figure 10). Forests and shrubland, which are widely distributed in the northern part of the Da Hinggan Ling Prefecture and the Xiao Hinggan Range, are relatively well-preserved and comprise vast virgin forests in China, including the Xing'an larch, sylvestris pine and white birch, etc. These species are usually rich in oil and are easy to burn. Due to the dry spring wind in the northeast region, most of the fires are attributed to natural disasters in Northeast China, e.g., lightning strikes. Therefore, the prevention and control of forest fires in Northeast China are important prerequisites for forest resources conservation and ecological security, as well as for the stability of production and life. In addition, the surrounding areas of the provincial capital cities of Harbin, Changchun and Shenyang in Northeast China are mainly induced by the fire-induced Stlmt-pattern. In fact, from 2012 to 2020, fires in areas with intensive human activities were rare in Northeast China. The fire-induced other-pattern is only sporadically distributed in the Songnen Plain and its southern regions, and this pattern is located around the fire-induced Agri-pattern. Because there is little vegetation cover in the unused land and it is also related to the centralized burning of straw in different places at local scale, further research is needed in the near future with finer-resolution data.

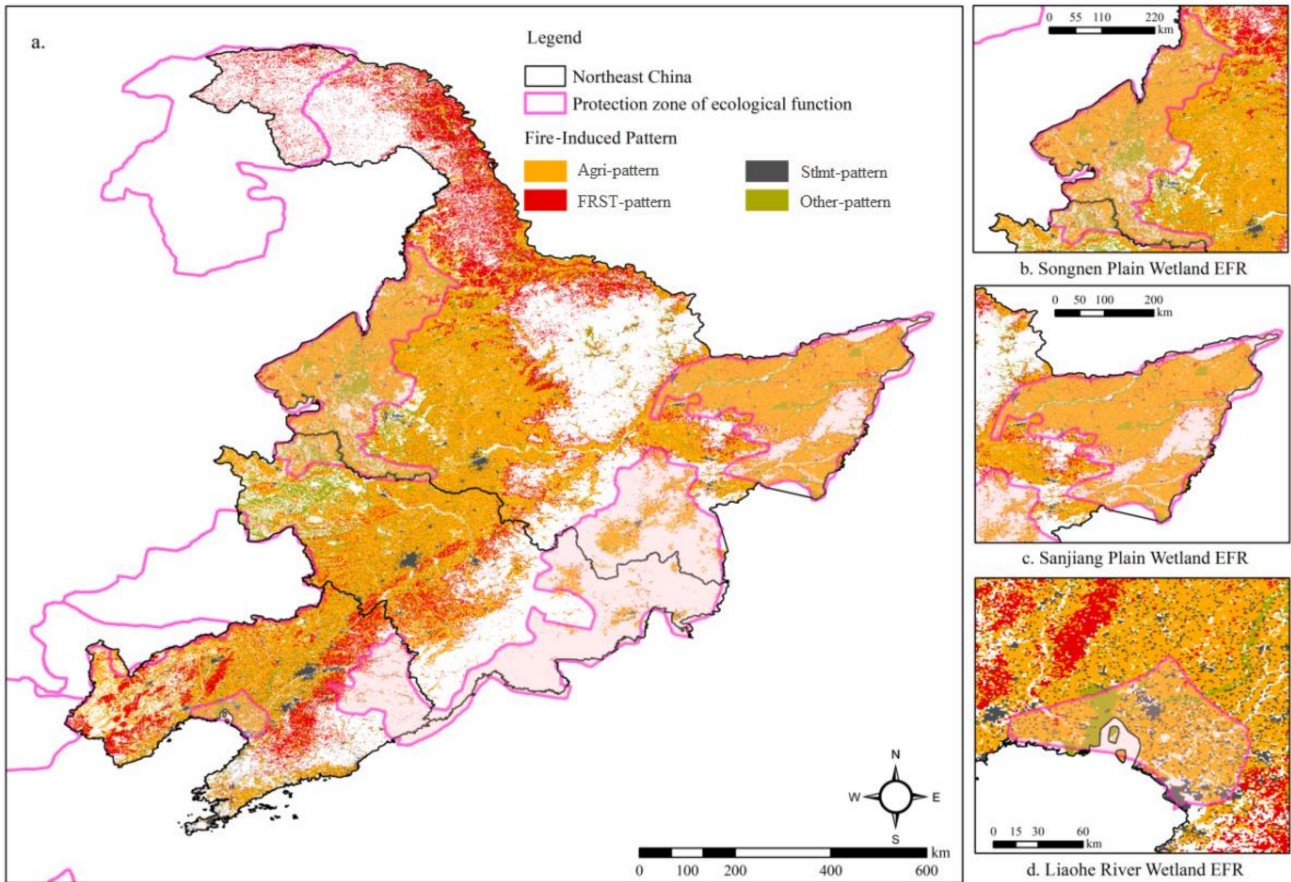

**Figure 10.** (**a**) Four types of fire-induced pattern in Northeast China. Note: (**b**) the Songnen Plain Wetland EFR, (**c**) the Sanjiang Plain Wetland EFR and (**d**) the Liaohe River Wetland EFR.

## 4. Discussion

Due to its unique geographical location, Northeast China has formed a unique climate with rising ground temperatures and strong evaporation in spring. When the rainy season has not yet arrived, the air is dry and warm [24,25]. The favorable soil and moisture environment provides conditions for the growth of bulk crops, and also retains a vast area of virgin forests. Under the influence of human activities and natural factors, agricultural and forest fires occur frequently in Northeast China. At present, straw smashing is not suitable for the northeastern region, and it may bring the risk of pests and diseases, while the in situ burning of crop residues is economical and effective [26,27]. In recent years, China has strengthened the control of man-made active incineration to achieve the goal of energy saving and emission reduction, but the environmental impact of straw burning is still unclear. These issues need to be further studied in order to find a balance between farmers' livelihoods and emission reduction policies [28,29]. Meanwhile, fire prevention and measurement are increasingly strengthened. However, fire management and forecasting are still relatively weak. Therefore, it is necessary to assess fire occurrence over a long historical period and finally determine the fire risk level of different spaces [30–32]. The fire occurrence levels and fire occurrence patterns in different areas in Northeast China determined in this paper are of great significance for targeted management of fire management and fire forecasting.

Next, scientific management is required for anthropogenic straw burning in extremely high-incidence areas. Studies have shown that a certain degree of straw burning in Northeast China is conducive to the accumulation of inorganic salts required for plant growth [33,34]. This primitive treatment process of crop residues maintains the ecological environment of the black earth in Northeast China to a certain extent. For forest fires in

very high-incidence areas, it is necessary to strengthen supervision and administration and open fire barriers. Regarding the large forest areas in the EFRs, it is still difficult to achieve timely results in man-made firefighting after forest fires. On the premise of not affecting the ecological function of nature reserves, opening up a fire barrier is an important way to prevent sudden forest fires [35,36]. For man-made fires in urban areas, the awareness of fire prevention during peak electricity consumption periods should be strengthened [37,38].

Finally, Northeast China has many trees rich in oil; it is easy to cause forest fires due to lightning strikes and in the dry and warm climate in spring. Many forest fires reported in Northeast China were caused by natural factors. By contrast, the risk of man-made forest fires is greatly reduced under the implementation of China's strict fire prevention policies. In this study, some active fire-hazardous areas are located in ecological function reserves, which are rich in biological resources. Whether the occurrence of active fires has an impact on the function of ecological reserves will be further explored. In the near future, we will combine the specific ecosystem service functions of different ecological functional areas and the intensity of active fires in the EFRs during different time periods to evaluate the impact of fire occurrence on ecosystem services. By doing so, we aim to provide a research basis for fire prevention and ecological governance in EFRs.

## 5. Conclusions

Based on the occurrence frequency, FRP and spatial location attributes of active fire data, this paper constructs an FI index, which was further applied to classify and spatially visualize fire-hazardous areas in Northeast China during 2012–2020. Combined with terrain, land cover and NPP data, the characteristics of active fire occurrence were analyzed. Then, a conceptual decision tree model of fire-induced patterns was established. Finally, through the spatial visualization of GIS, a spatial map of the fire-induced patterns is formed. From 2012 to 2020, active fires in Northeast China were clustered in the southwest of the Sanjiang Plain, the east of the Songnen Plain, and the Liaohe Plain. The FRP shows a general increasing trend with a cycle of three years. Meanwhile, the accidental area, frequent area, prone area and high-incidence area accounted for 31.62%, 30.97%, 26.23% and 11.18% of total active fires, respectively. More importantly, active fires still occur more frequently in the Songnen Plain, the Sanjiang Plain, the Changbai Mountain and the Liaohe Delta Ecological Reserves. Finally, the occurrence of active fires in Northeast China has notable characteristics in terrain and land cover, and has a certain correlation with NPP. More than 90% of active fires occurred in geographical spaces with altitude <350 m, slope <3°, and NPP ranging 2500–5000 kg·C/m$^2$. Active fire occurrence is more easily distributed towards half-shady and sunny slopes and more than 75% of them are concentrated in croplands and forests.

**Author Contributions:** This work is the result of collaboration among W.L., P.L. and Z.F. All authors have equally contributed, reviewed and improved the manuscript. All authors have revised the final manuscript. All authors have read and agreed to the published version of the manuscript.

**Funding:** This study was supported by the National Natural Science Foundation of China, Grant/Award Numbers: 41971242 and 42130508, and the Youth Innovation Promotion Association of the Chinese Academy of Sciences, Grant/Award Number: 2020055.

**Institutional Review Board Statement:** Not applicable.

**Informed Consent Statement:** Not applicable.

**Data Availability Statement:** Some or all data and models that support the findings of this study are available from the corresponding author upon reasonable request.

**Conflicts of Interest:** The authors declare no conflict of interest.

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
