# Peer review of "Delineating Fire-Hazardous Areas and Fire-Induced Patterns Based on Visible Infrared Imaging Radiometer Suite (VIIRS) Active Fires in Northeast China"

_remotesensing, doi:10.3390/rs14205115_

Round 1
Reviewer 1 Report
The article deals with the monitoring and prediction of the occurrence of fires. This topic is extremely topical, also considering the large number of fires currently burning. These fires reach enormous extents and the damage is enormous. They threaten forests, agricultural crops, but also human dwellings and people's lives. From this point of view, I consider the solution of this work and the application of the obtained results to real use to be very important.
Locating zones with high levels of fire danger is also important. Even clarifying the origin of the fire plays a very important role in the prevention of further fires.
In the beginning of the article, the authors provide an overview of solved similar works and the motivation for creating this work with relevant references.
This work uses the Fire Intensity Index (FI), which was developed with the active fires of the Visible Infrared Imaging Radiometer Suite (VIIRS) and then applied to identify fire hazard areas in northeastern China. Furthermore, a conceptual decision tree model was constructed to delineate fire-induced patterns influenced by various factors in Northeast China.
The results of this work show the potential of this work for the mentioned applications.
Technically, the article is prepared at an excellent level and there are only formal errors in it that need to be removed.
Comments:
Throughout the article, a non-uniform style of writing values ​​and units of individual quantities is used. Somewhere there are spaces between values ​​and units, and somewhere they are not. With compound units, the use of spaces is also random. Please unify it according to the article template.
Quantities in equation (1) should be written in italic style. In the same way, in other parts of the article, the quantities should be in italics. For example, FRP and FD are quantities and not abbreviations. It should be written in italics if it is a scalar quantity.
References to the literature should also be separated from the text by spaces and not as in the article: "public health[4-6]." Why isn't there a space?
The discussion and conclusion are sufficient, but I did not find plans for further future research.
Why are the names of the journals in the references written in capital letters. Please check the reference writing style according to the template. If I'm not mistaken, it's different in the template.
Author Response
- The article deals with the monitoring and prediction of the occurrence of fires. This topic is extremely topical, also considering the large number of fires currently burning. These fires reach enormous extents and the damage is enormous. They threaten forests, agricultural crops, but also human dwellings and people's lives. From this point of view, I consider the solution of this work and the application of the obtained results to real use to be very important.
Reply: The authors thank the professional reviewer for his or her affirmative comments on the research topic and content as well as the potential meanings.
Locating zones with high levels of fire danger is also important. Even clarifying the origin of the fire plays a very important role in the prevention of further fires.
Reply: Agreed. Many thanks for your professional advice. The delineation of zones with varied levels of fire danger is one of the key research contents. In addition, the causes of fire were discussed in this paper.
- In the beginning of the article, the authors provide an overview of solved similar works and the motivation for creating this work with relevant references
Reply: We are very grateful to the reviewer for your affirmation of the review at the beginning of the article. Some necessary references and discussion related to the research topic were enriched in the revision as follows (Line 61-77).
“Internationally, according to the International Wildland-Urban Interface Code, the distance to the structural location of different fire risk points (such as gas stations, factories producing inflammables, etc.) is grouped into different levels of fire hazardous zones [18]. Correspondingly, a series of fire protection standards for different hazardous areas have been formulated [19]. In particular, after the San Bernardino fire in 1980, public resource regulations was passed in California and generated a thematic map of fire hazard severe area [20], including three levels of medium, high, and extremely high. In addition, other factors such as topography, climate, and population are also considered for generation of fire hazard maps, and the FHZ are finished according to the roles and contribution degree of the factors [21]. However, there is still a lack of classification of fire hazardous levels based on fire occurrence scenarios over long period, as well as the intensity and frequency of fire occurrences. The occurrence of active fire provides information such as the accumulation of combustibles and the size of the fire, etc. Therefore, the spatial expression of fire intensity is of great significance to the formulation of fire management measures such as key prevention in regional fire-hazardous areas.”
- This work uses the Fire Intensity Index (FI), which was developed with the active fires of the Visible Infrared Imaging Radiometer Suite (VIIRS) and then applied to identify fire hazard areas in northeastern China. Furthermore, a conceptual decision tree model was constructed to delineate fire-induced patterns influenced by various factors in Northeast China.
Reply: Agreed.
- The results of this work show the potential of this work for the mentioned applications.
Reply: Agreed. We are very grateful to the expert for your affirmation of the research results of this paper. Your affirmation is of great significance for our ongoing research.
Technically, the article is prepared at an excellent level and there are only formal errors in it that need to be removed.
Reply: Agreed. We are very grateful to the expert for your careful review. The authors have revised the full manuscript carefully.
- Comments:
Throughout the article, a non-uniform style of writing values and units of individual quantities is used. Somewhere there are spaces between values and units, and somewhere they are not. With compound units, the use of spaces is also random. Please unify it according to the article template
Reply: Agreed. We are very grateful to the expert for your careful review. We added the necessary space between numbers and units in accordance with journal standards and requirements.
For example:
“More than 90% of active fires occurred in areas with altitude <350 m asl, slope <3°, and NPP between 2500~5000 kg·C/m2.” Line 24-25
- Quantities in equation (1) should be written in italic style. In the same way, in other parts of the article, the quantities should be in italics. For example, FRP and FD are quantities and not abbreviations. It should be written in italics if it is a scalar quantity
Reply: Agree. The style of the equation was updated. In the text, FRP and FD are abbreviations for the referential variables, which are not italicized according to the journal format.
- References to the literature should also be separated from the text by spaces and not as in the article: "public health[4-6]." Why isn't there a space?
Reply: Agreed and revised accordingly.
- The discussion and conclusion are sufficient, but I did not find plans for further future research
Reply: Agreed. Many thanks to the expert for your affirmation of the Discussion and Conclusion sections. Future research plans or ideas were added in the updated version.
Please see Line 502-506 in Page 15.
“In the near future, we will combine the specific ecosystem service functions of different ecological functional areas and the intensity of active fires in the EFAs in different time periods to evaluate the impact of fire occurrence on ecosystem services. By do so, it provides a research basis for fire prevention and ecological governance in ecological function reserves.”
- Why are the names of the journals in the references written in capital letters. Please check the reference writing style according to the template. If I'm not mistaken, it's different in the template
Reply: Agreed. Thanks to the reviewer for your careful review. We have carefully revised the references section according to the journal requirements
Thank you for your efforts to improve our manuscript. We hope that our responses and the resulting changes will be acceptable. And we will be happy to work with you to resolve any remaining issues.

Reviewer 2 Report
In this manuscript the authors have tried to relate an VIIRs satellite based active fire index to other biophysical variables.
The paper fundamentally lacks an explanation on why well known Fire regime parameters inlcuding Fire frequency/ duration/ spread rate, power flux, Total fire radiative power , parameters of the FRP distribition etc were not used. Instead the authors use a Gridded FRP / Grided density and provide no further explanation on how this is estimated or represented. Without any info it is difficult to review this paper.
2.3.2. Fire Intensity Index Construction section does not have any information and needs substantial improvement before the paper can be reviewed
I hope the authors can articulate why this index is prefered, its motivation and then go about describing the results
Author Response
- In this manuscript the authors have tried to relate an VIIRs satellite based active fire index to other biophysical variables.
Reply: Thank you very much for the careful review of the content of this article. Your suggestions are of great significance to improve the quality of the article. At the same time, thank you very much for your professional comments and suggestions on the structure and content of this article. The article has been carefully revised according to your comments.
- The paper fundamentally lacks an explanation on why well known Fire regime parameters inlcuding Fire frequency/ duration/ spread rate, power flux, Total fire radiative power , parameters of the FRP distribition etc were not used. Instead the authors use a Gridded FRP / Grided density and provide no further explanation on how this is estimated or represented. Without any info it is difficult to review this paper.
Reply: We are very grateful for the professional comments. As mentioned by the expert, the original data of active fire points includes attributes such as the location, time, and fire radiation power, characteristics of active fire occurrence can be discussed comprehensively based on the above parameters through statistical methods.
Although FRP and other attributes of an active fire "spot" can be spatially explicit, they cannot effectively depict the combined effects of FRP and fire frequency simultaneously in the same site. At the same time, the total fire radiation frequency in a certain geographical space is high due to two major reasons. First, there are active fire events with very small FRP. Second, very few or even one large fire occurred in a certain area resulting to a high value of FRP. For this reason, the FI index is established to overcome the situation that the local total FRP is large due to the occurrence of multiple active fire events. Based on the GIS spatial analysis, the FI index aims to spatialize the frequency of active fire (including location information) and the FRP simultaneously.
- 3.2. Fire Intensity Index Construction section does not have any information and needs substantial improvement before the paper can be reviewed.I hope the authors can articulate why this index is prefered, its motivation and then go about describing the results
Reply: Agreed. This comment is similar to the second one. The FI index is a key research content in this manuscript. We explained and added the necessary description in the revised version.
Thank you for your efforts to improve our manuscript. We hope that our responses and the resulting changes will be acceptable. And we shall be pleased to work with you to resolve any remaining issues.

Reviewer 3 Report
This manuscript proposed a fire intensity index using the Visible Infrared Imaging Radiometer Suite (VIIRS) active fire products and applied it to identify fire-hazardous areas in terms of different fire conditions: accidental, frequent, prone, and high-incidence, in Northeast China. The authors concluded with quantitative statistical results, inferring an increase of active fires in agriculture, followed by small forest fires. There are structural issues and a number of grammatical errors that if addressed would increase the readability of the manuscript tremendously.
In the introduction section, how do the other studies tackle fire-hazardous areas, and what methods were utilized? Why the proposed fire intensity index was an appropriate indicator? The literature review was not comprehensive. Further, the methods, results, and discussion need to have better alignment, how does the fire intensity index can improve the understanding and management of fire occurrence?
Line 166, this is the first use of the point feature, the authors need to elaborate on the point.
Line 179, …spatializes the frequency of active fire (including location information) and the fire radiation power. How did the authors conduct spatialization?
Line 193-233, These are some of the quantitative descriptions of datasets.
Line 457-459, These are descriptions of the study area.
Author Response
- This manuscript proposed a fire intensity index using the Visible Infrared Imaging Radiometer Suite (VIIRS) active fire products and applied it to identify fire-hazardous areas in terms of different fire conditions: accidental, frequent, prone, and high-incidence, in Northeast China. The authors concluded with quantitative statistical results, inferring an increase of active fires in agriculture, followed by small forest fires. There are structural issues and a number of grammatical errors that if addressed would increase the readability of the manuscript tremendously
Reply: Agreed. Regarding the structural issues and grammatical errors, the authors carefully revised the full manuscript to improve the readability. Since there are many revisions on the language, we suggest the referee re-checking them in the updated version.
- In the introduction section, how do the other studies tackle fire-hazardous areas, and what methods were utilized? Why the proposed fire intensity index was an appropriate indicator? The literature review was not comprehensive. Further, the methods, results, and discussion need to have better alignment, how does the fire intensity index can improve the understanding and management of fire occurrence?
Reply: Agreed. Your suggestions are of great help to the improvement of the paper. We added the review about the previous studies on the methods of mapping fire hazardous areas. Internationally, according to the International Wildland-Urban Interface Code, the distance from the structural location of different fire danger points (such as gas stations, factories producing inflammables, etc.) is divided into different levels of fire hazardous zones. Correspondingly, a series of fire protection standards for different hazardous areas have been formulated. In particular, after the San Bernardino fire in 1980, California passed public resource regulations in its legislation, which developed and drew a map of the fire hazard severe area in the area, which is divided into three types of hazard areas: medium, high, and extremely high. In addition, other factors such as topography, climate, and population are also considered in other fire hazard maps, and the FHZ are divided according to the influence degree of the factors.
However, there is still a lack of classification of fire hazardous levels based on fire occurrence scenarios in a long historical period, as well as the intensity and frequency of fire occurrences. The occurrence of a fire implies information such as the accumulation of combustibles, the size of the fire, etc, Therefore, the spatial expression of fire intensity is of great significance to the formulation of fire management measures such as key prevention in regional fire-hazardous areas.
By the way, we made necessary adjustments and/or revisions about the methods, results, and discussion. See revised manuscript for details.
- Line 166, this is the first use of the point feature, the authors need to elaborate on the point.
Reply: Thank you for your careful review and professional opinions, and the point density analysis method is detailed in the manuscript as follows.
“The Point density analysis tool can be used to calculate the density of point features around each output raster cell. Conceptually, a neighborhood is defined around the center of each raster cell, and the number of points in the neighborhood is summed and divided by the neighborhood area to derive the density of point features. If a value of each item (other than NONE) is used for the Population field setting, it will be used to determine how many times the point are counted. Calculation of the magnitude within each unit area is generally based on the point features that fall into the neighborhood around each unit. The larger the value of the input radius, the higher the generalization degree of the generated density grid will be, and vice versa. When calculating the density, only the points that fall within the neighborhood are considered. If no point falls within the neighborhood of a specific cell, then NoData is assigned. In this study, the spatial resolution of the output raster is 500m, with the search shape as circle. By contrast, the search radius is 100 active fire "points" centered on a point. It should be noted that one raster represents multiple active fire events that occurred on the same site in different time periods. Finally, the point density analysis tool was applied to conduct grid spatialization of active fire occurrence frequency and fire radiation power (FRP).” Line 189-204
- Line 179, …spatializes the frequency of active fire (including location information) and the fire radiation power. How did the authors conduct spatialization?
Reply: Thanks to the reviewer for your careful review. The spatialization of active fire occurrence frequency and fire radiation power in this paper is based on the point density analysis tool.
“It should be noted that one raster represents multiple active fire events that occurred on the same site in different time periods. Finally, the point density analysis tool was applied to conduct grid spatialization of active fire occurrence frequency and fire radiation power (FRP).” Line 201-204
- Line 193-233, These are some of the quantitative descriptions of datasets
Reply: Thank you for your careful review and professional advice. The original data of active fire is a number of vector points scatter geographically. Actually, many "points" of active fire overlap in space, which cannot intuitively tell the frequency of active fire in different areas. Likewise, each active fire "spot" indeed contains the attribute of fire radiant power, which can be computed statistically. However, the spatial statistics of fire radiation power through the point density analysis tool is helpful to understand the spatial patterns of fire intensity. By the way, Line 193-233 are based on the processing results of point density analysis. The results are necessary for the identification of active fire hazardous areas in the following sections.
- Line 457-459, These are descriptions of the study area
Reply: Agreed. Thank you for your careful review and professional advice. The content of line 457-459 was moved to the section of study area.
Thank you for your efforts to improve our manuscript. We hope that our responses and the resulting changes will be acceptable. We shall be happy to respond any further comments in the near future.

Reviewer 4 Report
The submitted research presents a fire intensity index with Visible Infrared Imaging Radiometer Suite (VIIRS) active fires. The proposed approach is then applied to identify fire-hazardous areas in Northeast China.
Generally speaking, I found the study very practical, interesting, and to the point. The manuscript is well-written, but there are some typos and language mistakes here and there.
The proposed method sounds simple and straightforward but also works well at least on the utilized sets of data. I will appreciate it if the authors a bit discuss the generalization of their proposed method on the other sets/types of data though. I would also encourage the authors to add a detailed discussion about the complexity of the proposed method in terms of computational time and so on. The abstract could be also modified to be shorter and concise.
Author Response
- The submitted research presents a fire intensity index with Visible Infrared Imaging Radiometer Suite (VIIRS) active fires. The proposed approach is then applied to identify fire-hazardous areas in Northeast China. Generally speaking, I found the study very practical, interesting, and to the point. The manuscript is well-written, but there are some typos and language mistakes here and there.
Reply: Thank you for your affirmation and encouragement. The proposed approach is useful and practical with current satellite-derived active fire data products. All the comments including the typos and language mistakes were carefully revised one by one. Please take a few minutes and recheck the seriously revised version.
- The proposed method sounds simple and straightforward but also works well at least on the utilized sets of data. I will appreciate it if the authors a bit discuss the generalization of their proposed method on the other sets/types of data though. I would also encourage the authors to add a detailed discussion about the complexity of the proposed method in terms of computational time and so on. The abstract could be also modified to be shorter and concise.
Reply: Firstly, the abstract was shortened and made it more concise accordingly.
Secondly, with the available satellite-derived active fire point products, including those from the Moderate Resolution Imaging Spectroradiometer (MODIS) and Visible Infrared Imaging Radiometer Suite (VIIRS), the proposed method can be introduced across the world.
Thirdly, regarding to the computational time and efficiency, to our best knowledge, the GIS-based analyses at regional, continental to global scales may take several hours to several days, which highly depends on the performance of the computer. Anyway, the computation is acceptable for large-scale analysis.
First of all, this paper establishes the exponential relationship between fire radiation intensity and occurrence frequency. The establishment of the FI is helpful to trace the characteristics of active fire occurrence in different geographical spaces. If the frequency feature is used alone, it is impossible to describe the intensity of each active fire event. Similarly, using the fire radiation frequency attribute feature alone, it is impossible to express whether this intensity is caused by a typical large-scale active fire over a period of time, or is caused by the superimposed intensity of multiple small-scale active fires over a period of time. Therefore, by using the fire intensity index and spatialization, it is possible to characterize the intensity of active fire in different geographical spaces. The above method can also be applied to the expression of urbanization process, for example, the residential area represents the residential agglomeration characteristics of local residents with vector point data, with combining the actual population, total population income, total electricity consumption, grain output and other data of the settlement, it can reveal the changes in the quality of life in the transition from rural to urbanization.
In addition, this paper also establishes a spatial pattern recognition based on the characteristics of active fire occurrence. Its essence is the logical classification method of decision tree, which is widely used and can be realized with the help of different data sets in aspects such as spatial pattern recognition and partition classification.
